# "Everything Is Old": National Socialism and the Weathering of the Jews of Łódź

Elizabeth Strauss 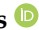

History Department, Mount St. Mary's University, 16300 Old Emmitsburg Rd., Emmitsburg, MD 21727, USA; estrauss@msmary.edu

**Abstract:** Using the social scientific theory of "weathering", the case study presented here reveals the broader explanatory power of the theory. Arline Geronimus developed the concept to describe the impact of racist systems on marginalized populations. Based on more than four decades of empirical research, Geronimus posits that the cumulative impact of navigating the structural racism embedded in US institutions results in accelerated declines in health and premature aging. The historical case study of the Łódź ghetto demonstrates that Nazi persecution of the Jews during the Holocaust resulted in a similar process of weathering among Jews. From 1939 to 1945, German authorities systematically dispossessed and uprooted, purposely starved, and exploited for labor the tens of thousands of Jews held captive in the Łódź ghetto. Despite valiant Jewish efforts to ameliorate the hardships of life in the ghetto, the persistent onslaught of racist policies and degradation ultimately resulted in widespread weathering of the population on an individual and communal level. I propose that the concept of "weathering" developed by social scientists has broad interpretative power for understanding the personal and communal impact of white supremacist societies in a historical context. The case of the Łódź ghetto is instructive beyond what it reveals about the particular persecution of the Jews during the Third Reich. The abrupt imposition of a racist system of government, the steady escalation of antisemitic policies from oppression and exploitation to genocide, and the relatively short duration of the ghetto's existence lays bare the cumulative effects of widespread individual weathering on the vitality of the community itself. In the Łódź ghetto, prolonged exposure to an environment governed by white supremacy also resulted in communal weathering.

**Keywords:** National Socialism; weathering; Holocaust; antisemitism

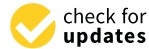



## 1. "Everything Is Old": National Socialism and the Weathering of the Jews of Łódź

> Everything is old, mummy-like, as in a herbarium; people, houses, trees, memories. Even fresh vegetables, as soon as they are brought into the ghetto, shrivel, the leaves shrink, the little tails of radishes and turnips wilt. . .Everything looks as in a secondhand shop, smelling like old *tefillin* [phylacteries] and prayer shawls . . . (Rosenfeld 1994, pp. 84–85)
>
> ~Oskar Rosenfeld, Łódź Ghetto

Oskar Rosenfeld recorded these words in his diary in February 1942, only three months after his incarceration in Łódź ghetto, the second largest Jewish ghetto in Nazi-occupied Poland. In the fall of 1941, Rosenfeld and nearly 20,000 other Jews were forced from their homes in cities across the German Reich and transported to the Jewish ghetto in the German-occupied Polish city of Łódź. The German-speaking Western Jews reluctantly joined nearly 150,000 native Polish Jews in the impoverished neighborhood of Bałuty. German authorities had chosen Bałuty as the site for the Jewish ghetto due to its reputation for squalor. Like most ghetto inhabitants, it did not take Rosenfeld long to recognize the deleterious effects of the ghetto. Everyone and everything withered and aged.

Adolf Hitler's regime engaged in a violent campaign between 1939 and 1945 to remake Europe in the image of National Socialist ideology. The outcome of Hitler's war

and genocide is well known. Approximately six million Jews and five million others were murdered in the name of Nazi white supremacy. These statistics offer some cursory insights into the impact of National Socialism—the incomprehensible number of victims, the vast geographical scope, the cooperation of non-German collaborators, and the apathy of countless bystanders. But what was the impact of the National Socialist brand of racism beyond the staggering number of people murdered at the hands of the regime? Hundreds of thousands of Jews lived in German-controlled territory for years before gas chambers and killing centers were employed as the "Final Solution to the Jewish Question" in 1942. How did German policies impact the victims of Nazi persecution prior to mass murder? How did Jews navigate the inherently racist system in the ghetto?

In the months following Germany's swift defeat of Poland in September 1939, ghettoization emerged as the Nazi regime's preferred method for subjugating local Jewish populations in the newly occupied territory. Among the thousands of Jewish ghettos erected under the auspices of the regime, the ghetto in Łódź operated the longest. The longevity, demographic make-up, and operation of the Łódź ghetto make it an ideal case study for an examination of the impact of structural racism on the marginalized. From May 1940 to January 1945, approximately 210,000 Jews lived in the ghetto. Only 7000 survived to witness the defeat of the Reich in May 1945. Nazi antisemitism permeated every aspect of the ghetto system, and German terror acted as a weapon of subjugation. As a result of prolonged, relentless exposure to structural racism and violence, individual Jews and the ghetto community as a whole succumbed to a process of "weathering".

The case study that follows builds on three critical academic contexts that have emerged from recent scholarship in the social sciences and humanities. First, central to my argument is the concept of "weathering". Dr. Arline Geronimus, professor of Behavioral Health and Health Education at the University of Michigan's School of Public Health, proposed the concept of "weathering" to explain why people of color in the United States experience deteriorating health at earlier stages of the life course and premature aging at higher rates than white Americans. According to Geronimus, "weathering" is "a process that encompasses the physiological effects of living in marginalized communities that bear the brunt of racial, ethnic, religious, and class discrimination [. . .]. Weathering afflicts human bodies—all the way down to the cellular level—as they grow, develop, and age in a racist, classist society [. . .]. Weathering is about hopeful, hardworking, responsible, skilled, and resilient people dying from the physical toll of constant stress on their bodies, paying with their health because they live in a rigged, degrading, and exploitative system" (Geronimus 2023, pp. 20–21). Human bodies naturally produce stress hormones in response to instances of oppression. Continual exposure to such instances of racial oppression results in prolonged elevated stress levels and has detrimental effects on the body, including muscle atrophy, immunosuppression, and premature aging. Geronimus' research suggests that while the broad spectrum of coping mechanisms exact a protective function for individuals in the short term, in the long term, such strategies ultimately hasten physical senescence (Geronimus 1992). When Geronimus first introduced her hypothesis in 1992, challenging traditionally accepted explanations for observable deterioration among marginalized populations, it was met with skepticism. However, subsequent research has produced compelling evidence in support of the theory.[1] Historical evidence from the Łódź ghetto lends further credence to the idea that prolonged exposure to a racist society and violence produces a weathering effect in individuals. Widespread individual weathering in the Łódź ghetto, particularly after 1941, culminated in a process of collective weathering that decimated the Jewish community. The historical case study presented below suggests that a broader application of weathering theory could be instructive for understanding the effects of racism on marginalized communities in contemporary contexts.

Second, my analysis of the weathering effects of Nazi persecution on the Jews of Łódź has been shaped by emergent and well-established trends in scholarship on the Holocaust. I am continually indebted to the scholars in the field who, with great care and clarity, have documented the intrinsic connection between individual and communal experiences

during the Holocaust (See, for example, Porat 1998; Ofer and Weitzman 1999; Beinfeld 1998; Shapiro 1999; Deletant 2004; Corni 2002; Sterling 2005). Growing interest in the experiences of the aged and aging victims of Nazi persecution has generated compelling studies that restored agency to an overlooked demographic and demonstrated the importance of age as an instructive category of historical analysis.[2] The topic and context of this article build on my previous work on elderly Jews in the Łódź ghetto, but the argument represents new thinking in order to (1) elucidate the connection between individual and collective Jewish experiences during the Holocaust and (2) demonstrate the potential power of a multidisciplinary approach.[3]

Third, my investigation of weathering in the context of the Łódź ghetto builds scholarship that places the persecution of Jews during the Holocaust and the oppression of people of color in the United States in a comparative context. While the case study presented here is not fundamentally comparative in nature, my work has been shaped by two important trends in this field of study. On the one hand, historians and legal scholars have extensively documented the convergence of racist ideas and their practical implementation in the United States and Germany in the first half of the twentieth century.[4] On the other hand, scholars rightfully warn against comparisons between the experiences of Black Americans in the US and Jews in the Third Reich that are overly simplistic, misleading, or exploitative (See for example Brown and Christmas 2014; Christmas 2015). Although the analysis that follows highlights similarities in the degrading cumulative effects of living in a racialized society, I acknowledge important ideological and practical differences between Jewish experiences during the Holocaust and the effect of American racism on marginalized groups. The speed and genocidal intent of Nazi persecution set it apart from other historical and contemporary contexts. This important difference does not detract from instructive parallels. Rather, the intensity of Nazi persecution of the Jews brings the weathering effects of racist societies into sharper focus.

On 1 May 1940, local German authorities in Łódź erected a fence around the border of the Jewish ghetto and handed down strict orders to members of the German *Schutzpolizei* [*Schupo*] tasked with guarding the perimeter to shoot anyone who approached the fence from inside the ghetto. Within the confines of these borders, a new social order crystallized— one rooted in a Nazi ideology that deemed Jews "racially inferior". For the next four years, German authorities dispossessed and uprooted, purposely starved, and exploited for labor the Jews held captive in the Łódź ghetto. Despite valiant Jewish efforts to ameliorate the hardships of life in the ghetto, the persistent onslaught of racist policies and degradation ultimately resulted in widespread weathering of the population on an individual and communal level. I propose that the concept of "weathering" developed by social scientists has broad interpretative power for understanding the personal and communal impact of white supremacist societies in a historical context. The case of the Łódź ghetto is instructive beyond what it reveals about the particular persecution of the Jews during the Third Reich. The immediate imposition of a racist system of government, the steady escalation of antisemitic policies from oppression and exploitation to genocide, and the relatively short duration of the ghetto's existence lays bare the cumulative effects of widespread individual weathering on the vitality of the community itself. I argue that prolonged exposure to an environment governed by white supremacy also resulted in communal weathering.

## 2. Ghettoization: The Imposition of Nazi Rule

Despite extensive preparations for war and the expansion of racist laws prior to the invasion of Poland, Hitler's regime had no clear, coordinated plan for the Polish Jews. After months of infighting between Nazi leaders on the ground and in Berlin, by the end of 1939, a consensus emerged in favor of concentrating and isolating Jews from the surrounding population.[5] In a secret decree from 10 December 1939, the Governor of the Łódź District, Friedrich Übelhör, outlined plans for a separate Jewish residential area. On 8 February 1940, German Police Chief Johannes Schäfer issued a decree that officially established the

ghetto and ordered Jews to move to the prescribed space located in the neighborhoods of Bałuty and Old Town, northeast of the city center.[6]

While a sizable portion of the Jews of Łódź resided in and around Bałuty prior to Germany's invasion and occupation of the city, complete isolation of the Jewish community required a massive population shift. Jewish reactions to German orders for ghettoization in February 1940 were mixed. Some embraced the resettlement. For those Jews who suffered routine abuse and humiliation from German authorities in the early weeks of the occupation, the ghetto had a protective appeal. Yes, the price of ghettoization was steep—dispossession, subjugation, and isolation. At the same time, within the walls of the ghetto, Jews, at least initially, no longer lived in fear of encountering their German persecutors face-to-face. Many Jews were reluctant to abandon their homes, possessions, neighbors, and daily routines. The process of ghettoization was slow to start. Frustrated with the pace of Jewish resettlement, German authorities utilized familiar tactics of enforcement: terror and violence.

During the first week of March 1940, German authorities terrorized the Jewish population in a violent action intended to complete the process of Jewish concentration and isolation in the ghetto, to showcase German power and Jewish subordination, and to instill fear in ghetto residents. The night of 7–8 March became known as "Bloody Thursday" after German police and SS officers murdered hundreds of Jews on the street and in their homes as they chased the Jews of Łódź into the impoverished neighborhood of Baluty in the northwestern corner of the city. Contemporary descriptions painted an unsettling scene of a faceless mass of disoriented Jews stumbling toward the ghetto, stooped under the weight of their possessions.[7] In his postwar memoirs, Łódź Ghetto survivor Itzhak Cytrynowski described the impact of the German terror that night: "After that night's events, there was no longer any need for encouragement to make us go and live in the ghetto. It was no longer a question of what we could save or not; rather it had become a life-or-death priority to obtain a roof over our heads in the newly designated area" (Cytrynowski 1988, p. 226). For Cytrynowski, the violence and terror of "Bloody Thursday" revealed with greater clarity the existential nature of the Nazi threat and sharpened his instinct for survival.

While the frequency of German violence against the Jews of Łódź subsided in the months following "Bloody Thursday", the traumatic memories of that night and the pervasive threat of violence loomed over every corner of the ghetto. Soon after, German authorities instituted a series of ordinances that increasingly restricted the movement of ghetto residents beyond its borders. In the first week of May, German police erected a physical barrier between the ghetto and the surrounding city and outlawed all communication and interaction between Jews in the ghetto and Christian Poles living outside the ghetto walls. The "success" of Jewish isolation in Łódź was contingent on a regime of relentless terror. Members of the German *Schutzpolizei* [*Schupo*] guarded the perimeter of the ghetto and followed strict orders to shoot anyone who approached the fence from inside the ghetto. *Schupo* guards murdered dozens of Jews at the ghetto border before Łódź was liberated in 1945. For all the Jews of Łódź, the fence was an inescapable, daily reminder of their subjugation and precarity.

In addition to terror and violence, manufactured scarcity was an essential feature of German rule over the Jews of Łódź. In the absence of an explicit objective or a clear strategy for resolving the "Jewish problem" in occupied Poland, local German authorities took a "productionist" approach to the administration of the ghetto and its residents.[8] The German Ghetto Administration in Łódź, led by Hans Biebow, aimed to extract as much wealth from the ghetto as possible, including monetary assets, personal property, and forced labor for the regime. At the same time, occupation authorities instituted oppressive policies that restricted the quality and quantity of available foodstuffs and created a physical environment more conducive to the spread of disease. Through the end of 1941, extraction and exploitation remained the purpose of the ghetto. That the mortality rate among Jews increased tenfold over prewar rates in the first months of the ghetto's existence as a result of these policies was an added bonus for German authorities.

### 3. Weathering: The Cumulative Impact of the Nazi System

The Nazi regime had little interest in maintaining or managing the day-to-day operations of the Jewish ghetto. When the ghetto borders were closed in May 1940, German authorities bestowed the Jewish Council of Łódź with administrative responsibilities. The German Mayor of Łódź, Franz Schiffer, granted the Chairman of the Łódź ghetto Jewish Council, Mordechai Chaim Rumkowski, broad authority to manage the internal life of the ghetto.[9] In his capacity as Council chairman, Rumkowski was the official representative and spokesman of the Jewish community and liaison between German authorities and ghetto residents. Rumkowski capitalized on his relative freedom to consolidate his power and build a massive bureaucracy that could address the needs of thousands of ghetto inhabitants. The implementation, duration, and success of Jewish initiatives to stem the weathering effects of the Nazi system was wholly dependent on German support. Despite individual and collective efforts at amelioration, neither individual coping mechanisms nor collective measures could prevent the deleterious effects of daily depravation and injustice.

After the traumatic process of resettlement into the ghetto, extreme overcrowding and dilapidated accommodations across the ghetto area acted as another assault on the Jews of Łódź. The Jewish Council faced a population in crisis when the ghetto was sealed. In May 1940, the impoverished neighborhoods designated for the ghetto occupied 4.13 square kilometers in a northeastern section of Łódź. Available living quarters were crowded into 2.41 square kilometers. Approximately 160,000 Jews were crammed into 3361 one- and two-story buildings that housed 31,962 dwellings of one or two rooms. The vast majority (95 percent) of the dwellings had no hygienic facilities.[10] Running water, sewage pipes, and bathrooms were a luxury that only 49 out of the total dwellings enjoyed. The population density was about seven times the average in prewar Łódź.

Finding shelter was the most pressing concern for Jews forced into the Łódź ghetto. Some were taken in by family and friends who already resided within the ghetto boundaries. In other cases, Polish landlords extorted Jewish refugees, demanding hefty fees in exchange for keys needed to access vacant flats. Still others fought with Polish occupants who refused to leave their dwellings despite German demands. Amidst the chaos, some ghetto dwellings housed ten to fifteen people. The dramatic increase in population density for Jews in the ghetto acted as a psychological assault intended to unsettle and disorient new ghetto residents. As families shifted and moved between dwellings, the average number of people living in each ghetto dwelling settled at about three or four people per room.

Heightened stress levels provoked by crowded and uncomfortable living conditions were further exacerbated by the deplorable state of sanitation in the ghetto and prolonged hunger. Ghetto inhabitants were constantly on edge. Tensions frequently devolved into fights between loved ones and neighbors over food rations and personal space. Overcrowding and insufficient hygienic facilities promoted the rapid spread of disease. The cumulative effects of a multifaceted assault on the physical and psychological well-being of Jews in the ghetto suggest a process of accelerated weathering.

Hunger was the most pervasive and relentless threat to the well-being of Jews living in the Łódź ghetto. From the outset, hunger ravaged the Jews living in the second-largest ghetto in German-occupied Europe. Germans supplied the ghetto with very little food, and the hierarchical scheme for food distribution in the ghetto privileged the few who occupied elite positions in the administration. Widespread starvation made it difficult for families to maintain the level of civility necessary for the survival of the family unit. Extreme hunger was enough to degrade the strongest relationships. Elderly men and women were particularly vulnerable. Older Jews often relied on family members to collect and supply food. Many were not so fortunate to have a family member willing to help in this context. When older men and women were denied food by their families, they often turned to the Jewish Council for assistance. It must have been a crushing realization when commitments implied by the intimate bonds of family were not fulfilled.

In February 1941, 70-year-old Gitta Wajcenfeld requested aid from the community while living with her sons. She pleaded for help from "Mr. Rumkowski, who looks after

old people," implying that her family had failed to do so. Mrs. Wajcenfeld claimed that her sons left her "to starve completely".[11] In the wake of broken family commitments, she threw herself at the mercy of the Łódź ghetto welfare system.

Arguments over food could turn vitriolic, leading to family betrayals unimaginable before the ghetto. Older men and women living with younger family members found themselves in particularly precarious situations when it came time to dole out food rations. In April 1941, 65-year-old Jakov Blumenkrants described the corrosive effects of starvation on intergenerational relationships. Mr. Blumenkrants resided with his daughter, son-in-law, and their small children in flat number 10 at Litomierska 26 in the ghetto. During the first year of the ghetto period, while the family was still "earning a little something," he had enough to eat, and life was "tolerable". But in recent months, food had been scarce, and quarrels had erupted in the family.

Jakov Blumenkrants lamented the fact that everything was "naturally" taken out on him. The aging grandfather discovered that the children were stealing his food. Imagine the shock he felt when, upon protesting this injustice, his grandchildren insulted him and "spit [at him] in an inhumane manner". Family quarrels had become so disruptive that neighbors frequently intervened on the old man's behalf but to no avail. Feelings of betrayal and hopelessness pervaded each line of Mr. Blumenkrants' letter and were most evident in his admission that "I would rather be dead than stay with them in this flat".[12]

Blumenkrants' dilemma reflected the devastating effects of German policies. Hunger ravaged family relationships and pitted family members against one another. The construction of the ghetto ensured that instead of fighting the actual perpetrators of violence, Jews would struggle against one another to procure basic necessities for survival. In this context, elderly Jews fought with the children they had raised and the grandchildren they had loved. In the case of Mr. Blumenkrants, leaving the family unit offered a more tolerable solution than remaining isolated and shunned in a familial setting.

The family of 73-year-old Chanie Kotek appealed to admit the elderly woman to the community's home for the aged in March 1942. The family of eight simply could not tolerate Mrs. Kotek's belligerence, which seemed to intensify as she became weaker from hunger and illness. According to the petition submitted by her son-in-law, the old woman argued constantly with her grandchildren. The disruptive nature of Mrs. Kotek's interactions with family members and her worsening health motivated family members to remove the old woman from the family dwelling and place her in the care of the community.

In some cases, aged parents and grandparents felt that younger generations inhibited their chances for survival. After the death of his wife in July 1940, 76-year-old Chanania Chaim Engel moved in with his children.[13] A year later, in July 1941, Mr. Engel was ready to leave the family dwelling. A surface reading of his petition for admission to a home for the aged suggested that Engel believed himself to be a burden to the families. A closer examination of the letter revealed the possibility of additional motivations. For Engel, residence in the home for the aged was necessary because he "was extremely anxious" and required the "utmost quiet" to live.[14] What was Engel's intention? Did he hope to relieve his family of the undue burden of taking care of an aging man? Or did he wish to escape the chaos of the family home in the ghetto? Perhaps he sought refuge from a crowded and noisy apartment that threatened his mental and physical health.

As a result of these persecutory policies, ghetto residents experienced both physical and psychological accelerated weathering. Oskar Rosenfeld exposed the impact of the Nazi system on ghetto residents in a dialogue between the fictionalized ghetto inhabitants featured in one of Rosenfeld's surviving literary sketches entitled "Golem and Hunger". In a succinct and insightful monologue, Rosenfeld's character described the compounding nature of prolonged exposure to Nazi oppression:

> The end result of a process that began with expulsion from our homeland, depravation, and degradation of all rights. All is lost if hunger is joined to homesickness, fear, despair, or even weariness of life. An organism depleted by hunger cannot withstand such psychological assaults. This barren landscape, [. . .] a dreary cli-

mate, filth, dust, flies, excrement in yard and tree, all these plagues of ghetto life encroach on us day and night until we collapse. Enemies compass our bodies from all sides. We don't know which of the threatened parts we should protect first. We are constantly exposed to assaults on our heart, lungs, stomach, nerves. The muscles dwindle, and the bone frame becomes brittle. [. . .] (Rosenfeld 1994, p. 99)

In this poignant sketch, Rosenfeld made an explicit connection between the psychological and physical dimensions of weathering. Living in a constant state of fear made it even more difficult to resist physical decline.

Death became a routine part of daily life. And those left behind suffered from accelerated aging and failing health no matter their age. Łódź ghetto survivor Sarah Selver-Urbach remembered how her father's death in the fall of 1941 hastened the decline of her paternal grandfather. In a postwar memoir, Selver-Urbach wrote: "Following father's death, my tall, stately grandfather grew stooped; his hair, which had turned grey by father's grave, became whiter and whiter till his head and long beard looked snow-white. The permanent pain etched on his face augments further his naturally dignified appearance; it was sad to see him wasting away, he who had been so hale and so tough" (Selver-Urbach 1984, p. 75). The cumulative effects of starvation, fear, and death were manifest in the greying of the population.

In the Łódź ghetto, physical weathering emerged as a widespread phenomenon that often occurred at a shocking pace. Among the most vulnerable to accelerated aging were the 20,000 German-speaking Jews from cities in the Reich who were deported from their native lands and forcibly integrated into the Polish- and Yiddish-speaking native Jewish community of the ghetto. For the newcomers, the reality of occupied Poland was wholly disorienting. Still used to home-cooked food and other creature comforts from their former lives, in the beginning, they scoffed at the meager portions of thin, watery soup that passed as food in the ghetto. The deportees from the West quickly realized that the material conditions they had left behind in Berlin, Vienna, and Prague were far superior to those they faced in the Łódź ghetto.

Fellow residents of the ghetto observed that the rate of decline among "foreign" Jews outpaced even the most vulnerable within the native Polish community. In a report for the *Daily Chronicle*, a collective effort to document Nazi injustices against the Jews of Łódź spearheaded by Rumkoswki's administration, Jozef Klementynowski noted the speed with which Jews from Hamburg, Germany, succumbed to the process of weathering upon arrival in the ghetto. Klementynowski reported: "Events outpaced time; people changed visibly, at first outwardly, then physically, and finally, if they had not vanished altogether, they moved through the ghetto like ghosts. . . [. . .] And indeed, it was only half a year, only six months, that had proven to be an eternity for them! Some of the metamorphoses could not be imagined, even in a dream. . .Ghosts, skeletons with swollen faces and extremities, ragged and impoverished [. . .]" (Dobroszycki 1984, p. 166). Mortality rates among the "foreign" population in the ghetto corroborate anecdotal evidence.

In addition to physical weathering, ghetto residents also reported psychological impacts. Oskar Rosenfeld arrived in the Łódź ghetto in the fall of 1941, a spry and healthy 57-year-old. Within just a few months, Rosenfeld noted disturbing changes in his mental faculties in his diary:

I am myself in the grip of the most widespread ghetto disease: dimming of the memory. . .not being able to remember things just heard, the names just read. There is a flicker in front of the eyes, a drying in the ears, one hits one's forehead, racks one's brain, and attempts to conjure up the past. To no avail. (Rosenfeld 1994, p. 93)

"Ghetto disease" became shorthand for the brain fog experienced by so many ghetto inhabitants due to severe, prolonged caloric deficit. Insufficient caloric intake was just part of the problem. Foodstuffs in the ghetto lacked the most basic nutritional value. Jewish doctors working at medical facilities in the ghetto noted with dismay the emergence of "little-known or disregarded illnesses" in the malnourished ghetto inhabitants, including

scurvy, pellagra, and famine edema, which they linked to lack of essential vitamins and minerals in ghetto rations (Ibid., 177–178).

The highly structured and multifaceted nature of Nazi persecution created the ideal conditions for the rapid, unchecked spread of disease. Starvation weakened the bodies' natural defenses and hindered the healing process. Crowded living quarters facilitated the easy transmission of contagions between residents of the ghetto. In particular, outbreaks of tuberculosis and typhoid in the ghetto were major contributors to a sharp increase in mortality during the ghetto's existence. Between 1940 and 1944, the mortality rate in the ghetto averaged a staggering 21 percent.

### 4. Conclusions: Beyond Individual Weathering?

In the concluding discussion that follows, I propose that the theory of weathering has explanatory potential beyond the individual. Firsthand accounts from the Jews in the Łódź ghetto point to a phenomenon of collective weathering that mirrored the process of decline experienced by individual Jews living in the ghetto. Just as the cumulative impact of prolonged exposure to racist policies and practices in the ghetto initiated a process of accelerated weathering in individuals, the proliferation of individual weathering in all sectors of ghetto society and the violent assault on the Jewish community in Łódź in 1942 accelerated the decline of the Jewish community as a whole.

The natural sequence of life—that one generation follows the next—was turned on its head in the Łódź ghetto. That younger Jews died before the older generation became a devastating fact of life, especially after 1940. Mortality statistics from the first two years of the ghetto period showed that people over 60 years of age died at a faster rate than any other age group during the first months of the ghetto period. Older Jews accounted for approximately 45 percent of the total number of deaths between May and December 1940. Most often, the cause of death was registered as "heart disease" or "heart weakness" [*Herzkrankheiten* or *Herzschwäche*].[15]

In the summer of 1940, during the peak months of the dysentery epidemic that plagued the Łódź ghetto, nearly 45 percent of the total number of Jews who succumbed to the disease (2254 people) were over 60 years old.[16] A change occurred in 1941 and 1942. As the dysentery epidemic was brought under control, rates of elderly mortality slowed. At the same time, Jews in the age groups most likely to be conscripted into hard labor became increasingly vulnerable to the long-term effects of hunger and the spread of tuberculosis. As a result, in 1941 and 1942, there was a marked increase in death rates among ghetto residents aged 15 to 60 years.[17]

Widespread death among people considered to be in the prime of life had unsettling effects on the entire community. Oskar Rosenfeld remarked on this perversion of life's usual trajectory in his diary:

> The old saying "Die and become" was also given short shrift. The Jewish concept of *ovaus ovousaynu* [Hebr., *avot avotenu*; our forefathers] was likewise lost in the face of the daily events. In the past, generation followed generation in a natural sequence, in accordance with the historical consciousness, Jewish tradition, and biblical worldview. "He returned to his fathers" now everything is mixed up: the son before the father, the grandchild before the grandfather, the young before the old. Absurd, ludicrous, unharmonious—therefore, the chaotic states of mind; therefore, that which is called "godless". Dying has lost its celestial beauty; its wondrous secret felt sacrosanct. The mystery of death is desecrated hourly by the brutality of its earthly cause—hunger, which desecrates. (Rosenfeld 1994, p. 46)

The consequences of German policies—starvation, endemic disease, bodies worn down by hard labor, and ultimately, death—disrupted family dynamics and posed particular challenges for older Jews who outlived their younger family members.

In the summer of 1941, the Nazi regime initiated preparations for the mass murder of European Jewry on an industrial scale. The forced resettlement of nearly 20,000 mostly German-speaking "foreign" Jews to the Łódź ghetto in the fall of 1941 precipitated an even

more violent phase of Nazi persecution that escalated to full-scale genocide in 1942. As a collective, the newcomers, of whom more than half were older than fifty, challenged the status quo that had taken shape in the ghetto community since its inception in May 1940. The massive influx of people in an already overcrowded ghetto space caused terrible strain for the established community of natives as well as for the foreign Jews. Women from Berlin and Prague disembarked at the ghetto train station wearing fur coats and carrying luggage full of their most precious housewares. Most did not speak the languages of the ghetto—Polish and Yiddish—and could not easily communicate with their fellow ghetto captives. The native population was equally unprepared to accommodate such a large number of people, who appeared well-off in the midst of ghetto squalor and mostly spoke the language of the oppressor—German.

The accommodation of Jews from cities in the German Reich created new tensions within the ghetto community that weakened existing bonds and institutions. Due to common language and racist assumptions that Jews of German lineage were naturally better equipped to administer ghetto industry, German authorities elevated newcomers to positions of authority over native Polish Jews. The tactic of pitting native and foreign populations against one another ensured tensions between the populations remained high to the advantage of German oppressors. In order to accommodate thousands of new residents, Rumkowski made the difficult decision to transform ghetto schools into collective housing for the refugees. School closures compounded the weathering effects of prolonged, systematic oppression and scarcity and quashed hopes for the revitalization of the community. Abraham Cytryn captured the devastating effects in one of the short stories he recorded in his ghetto notebooks while incarcerated in Łódź. Describing the character Eli, a young boy navigating the hardships of ghetto life, Cytryn wrote: "Eli wonders, do our travails also obey the laws of nature? It's no wonder the boy poses such questions to himself. Life has made him old before his time. His soul inclines toward mysticism and does not permit logic to guide him. Often, his thoughts are not those of a child. His feelings are loftier than those of an adult[.] Unfortunately, there is no longer anyone who will see to his future education" (Cytryn 2005, p. 57). The genocidal policies of the Nazi regime and Jewish attempts to ameliorate further hardship ultimately weakened the community and stripped it of hope for a vital future.

Refugees continued to flood into the ghetto through the summer of 1942 as German authorities destroyed smaller ghettos of the *Wartheland* region, killing many people and sending those who remained to the ghetto in Łódź. The concentration of disparate Jewish communities in the Łódź ghetto in 1941 and 1942 facilitated Nazi genocidal policies and resulted in the murder of nearly 45 percent of the Jewish population in Łódź by the fall of 1942. In December 1941, German officials initiated operations at the killing center in Chełmno [*Kulmhof* in German], just thirty miles northwest of the ghetto. The commencement of mass murder in Chełmno ushered in a period of intense uncertainty, terror, and destruction for the Jews of Łódź. Between 16 January and 12 September 1942, more than 70,000 Jews were taken from the ghetto and murdered at the extermination facilities in Chełmno. Local German authorities implemented this series of "actions" [*Aktionen*] by co-opting Jewish institutions and demanding cooperation from the Jewish leadership. In January 1942, the Jewish administration was forced to arrange for the deportation of 10,000 ghetto inhabitants to their deaths at Chełmno. Deportation transports took more than 34,000 from the ghetto between 22 February and 2 April. Before a suspension of transports during the summer of 1942, close to 11,000 more Jews were taken to be murdered at Chełmno in the first half of May. These statistics represented more than individual lives lost; they also revealed a devastating blow to the community.

The transports of 1942 culminated in a final, devastating *Aktion* in September 1942 known as the *Shpere*.[18] Biebow demanded from Rumkowski a list of 20,000 names to include ghetto inhabitants under the age of ten and older than sixty-five. The violent week of 5–12 September left no family untouched and transformed the community overnight. German authorities removed from the ghetto and murdered two generations of people—killing

individuals on the spot, dismembering families, and dismantling vital social networks. The abrupt disappearance of people sixty-five and older, as well as children under ten, stripped the community of its living connection with the past and its potential for a future. Without the generational reference points of children and older adults, the remaining segment of the population aged overnight.

The effects of the *Aktion* were not just felt on an individual level but on the communal level as well. Families were destroyed in the process of the *Shpere*. In his memoir, Nirenberg noted the effect on families: "The Sperre left its mark on the survivors. This was not the first catastrophe, but the worst so far. Entire families had been killed, and the majority of survivors had lost at least one relative in the tragic Sperre days" (Nirenberg 2003, p. 64). Widespread trauma accelerated individual weathering and filtered out into the community. Communal activity virtually came to a halt after the *Shpere*, never to be the same again. After the *Shpere Aktion*, Oskar Rosenfeld observed that everything and everyone subjected to life in the ghetto shrivels and wilts. In the ghetto: "Everything is old" (Rosenfeld 1994, pp. 84–85).

In the immediate aftermath of building selections, survivors were forced to return to their dwellings and confront their losses. In the claustrophobic environment created by the curfew, the remaining Jews recognized the senescent effects of the *Shpere*. In Zelkowicz's words:

> When you sit in the room and wallow in your sorrow—your own and that of your wife and son—commiserating with your neighbors and all the Jews. . .when you have been sitting so long in that room, the mirror that you glanced at reflects an emaciated, fragile and crumbling figure. You, too, are a candidate for "throwing on the garbage heap. . .!" When you sit in the room and sneak a contemplative glance at your wife, who has aged many years in the past two days, and your handsome son, and you observe his joyless, harried features and the deathly terror emanating from his deep, dark eyes, the general fear leads you to dread for your lives: yours, your wife's, and your trembling son's. All of you are candidates; all of you have dim, withered, gray-green facial features. Those are the kinds of people they are seeking, the kind who are nabbed. (Zelkowicz 2002, p. 369)

Even the rare family left untouched by the *Shpere* succumbed to the effects of accelerated communal weathering.

The traumatic effects of the *Shpere*—the murder of two generations of the ghetto community—cast a pall over the ghetto for months and years after the round-ups ceased and the curfew was lifted. Zelkowicz described the ghetto as completely transformed after the *Shpere.* He wrote in his diary: "You know them well, your neighbors. The living conditions in the ghetto made sure of that. Today, however, you hardly know them at all. They have changed overnight—not only in their outer appearance but in their state of mind and inner essence [. . .] All your neighbors, all your acquaintances have been transformed inside and out, overnight" (Ibid., 369–370).

After September 1942, the ghetto effectively functioned as a work camp. The ghetto reached near full employment by 1943, with 85 to 95 percent of the remaining population working in the ghetto's workshops. Communal institutions closed. Social and cultural activities were minimal. Political groups, inasmuch as they still functioned, moved underground. Religious practice retreated from public spaces. The power that Rumkowski had consolidated over the previous two years into unchallenged authoritarian rule was undermined by his cooperation with the murderous German *Aktion*. German authorities further undermined his rule when they transferred control of major responsibilities to other ghetto leaders. After the *Shpere*, David Gertler, who was already head of the *Sonderabteilung*, assumed control of the supply distribution for the ghetto. Aaron Jakubowicz assumed responsibility for the *resorts*—the foundation of the ghetto industry. In addition to the dispersion of responsibilities among Jewish leaders, German authorities increased their direct involvement in the internal affairs of the ghetto.[19] If ghetto inhabitants had entertained any idea that the Jewish administration could provide aid and protection against Nazi

oppression, such notions died with tens of thousands of Jews from Łódź sent to the gas chambers in 1942. By the end of 1942, Nazi oppressions had transformed the once vibrant Jewish community of Łódź into a weakened group of weathered individuals without a functioning communal network.

An analysis of German persecution of Jews in the Łódź ghetto testifies to the explanatory power of weathering theory for elucidating the impact of living in a society governed by racist systems and practices. Manufactured depravation, persistent and pervasive fear, and ever-present threats of violence overwhelmed Jews in the Łódź ghetto and resulted in an accelerated individual decline. Regardless of important differences in timing and context, the hallmarks of Jewish weathering in the context of the Holocaust are remarkably similar to features of the weathering process in African Americans first documented by Arline Geronimus. Jewish reactions to the relentless onslaught of Nazi persecution in Łódź, documented in the discussion above, suggest that the rate and frequency of individual weathering was mirrored in the deterioration of the Jewish community as a whole. The phenomenon of communal weathering presented in the historical case study above parallels some recent findings about the wider impact of individual weathering on Black American communities in the United States.[20] Further research should aim to identify specific hallmarks of communal weathering and theorize the trajectory of its progression.

**Funding:** This research was funded by the United States Holocaust Memorial Museum, Barbara and Richard Rosenberg Fellowship; the Center for Jewish History, Dr. Sophie Bookhalter Fellowship in Jewish Culture, and YIVO Institute for Jewish Research, Maria Salit-Gitelson Tell Memorial Fellowship.

**Institutional Review Board Statement:** Not applicable.

**Informed Consent Statement:** Not applicable.

**Data Availability Statement:** Data are contained within the article.

**Conflicts of Interest:** The author declares no conflicts of interest.

## Notes

1.  Arline Geronimus' scholarship on the theory of weathering is extensive. Most recently Geronimus published a manuscript based on four decades of research on the weathering effects of structural racism in the United States. See (Geronimus 2001, 2023; Geronimus et al. 2006, 2019, 2020); For further support of the weathering hypothesis see Simons et al. (2021); Kammi Schemer's recently extended investigations of weathering to include minority children. See Schmeer and Tarrence (2018).

2.  See Hájková (2013, 2020); Löw et al. (2013). For more recent investigations of elderly experiences during the Holocaust see (Stone 2018; Raisch 2022).

3.  The first iteration of my research on the aged and aging in the Łódź ghetto was featured in my doctoral dissertation. See (Strauss 2014) More recently I examined survival strategies utilized by elderly Jews in the context of "survival through work"—the communal approach to survival in the Łódź ghetto. See Strauss (2021).

4.  In recent decades scholars from a variety of disciplines has explored the intellectual and legal connections between the United States and Nazi Germany. Prominent social scientists and doctors in the US and Germany promoted eugenics. See for example. Legal scholars and historians specifically have detailed how lawyers and bureaucrats for the Nazi regime drew inspiration from racist ideas that undergirded the foundation of the United States and codified racial segregation and oppression into law into the twentieth century. See (Miller 2020; Whitman 2017; Kakel 2011; Bachrach and Kuntz 2008; Kuhl 2002).

5.  Early anti-Jewish policies coupled with forced deportation to the *Generalgouvernement* were part of Greiser's plan to create a German city free of Jews. The resettlement of tens of thousands of ethnic Germans (*Volksdeutsche*), brought from territories in Soviet control *Heim ins Reich* starting in autumn 1939, intensified the pressure on the Jewish population of Łódź. During this period, thousands of Jews were sent to German-occupied eastern Poland, under the administration of Reich Governor Hans Frank. Only when Hans Frank objected to the influx of Jewish expellees into the territory under his control was Greiser forced to amend his plans for the Jews of Łódź. (Aly 1995).

6.  For an English translation of the decree, see Adelson and Lapides (1989). The ordinance included eight directions intended to guide the massive population transfers necessary to concentrate all of the Jews of the city into a single space. The original decree was brief and anticipated the need for more detailed instructions in the coming months.

7.  Ghetto resident Irena Liban provided an eyewitness account to this effect. See Adelson and Lapides (1989), pp. 34–35.

8.  For more detail about the regime's approach to the Jewish populations that came under German control in occupied Poland see Christopher Browning's discussion of the "attritionist" and "productionist" camps in Browning (1986).

[9] See a translation of the memorandum from City Commissioner Franz Schiffer to Rumkowski in Adelson and Lapides, *Łódź Ghetto*, 34–35, 47–48.

[10] Isaiah Trunk parsed the numbers of inhabitable and uninhabitable living space in the ghetto in his seminal work on the history of the Łódź Ghetto. See Trunk (2006).

[11] Gitta Wajcenfeld to Rumkowski, Petition, 18 February 1941. USHMM, RG-15.083M, File 267, Reel 90, 327.

[12] Jakov Blumenkrants to Rumkowski, Petition, 16 April 1941. USHMM, RG-15.083M, File 267, Reel 90, 594–595.

[13] Chanania Chaim Engel to Rumkowski, Petition, 21 July 1941. USHMM, RG-15.083M, File 267, Reel 90, 529–530.

[14] Ibid., 529. (Emphasis in the original).

[15] Department of Statistics, "Todesfälle nach Todesursachen im Jahre 1940"; Nachman Zonabend Collection; RG 241, Folder 800; YIVO.

[16] Department of Statistics, "Todesfälle nach Alter der Verstorbenen, 1940–1942"; Nachman Zonabed Collection; RG 241, Folder 800; YIVO.

[17] "Todesfälle nach Alter der Verstorbenen"; RG 241, Folder 800; YIVO. The number of deaths of people aged 15–60 in 1940 was 2492. Mortality rates of late adolescents and adults more than doubled in 1941 (6947 deaths) and remained at that increased level in 1942 (6647).

[18] The Yiddish designation for the *Aktion* came from the German word *Gehsperre*, meaning "curfew," referring to the German order that demanded the Jews of Łódź remain in their houses during the days of the *Aktion*.

[19] Isaiah Gutman, Introduction to *Lodz Ghetto* by Isaiah Turnk. See Trunk (2006), xl.

[20] See for example (Geronimus and Thompson 2004; Wacquant 1998).

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
