# Peer review of "“Everything Is Old”: National Socialism and the Weathering of the Jews of Łódź"

_genealogy, doi:10.3390/genealogy8020033_

Round 1
Reviewer 1 Report
Comments and Suggestions for Authors
Overall, this article drawing on the experiences the Jews forcefully concentrated into Łódź ghetto from 1939-1945, argues that the theory of weathering holds broader explanatory power that can help unpack the violence inherent in racialized structures, policies, and practices and that produces avoidable death through multiple pathways: physical health, premature aging, psychological distress, family breakdowns, and many more. The argument put forth by the author is very compelling. However, I think the paper can be strengthened to bring the paper to the level that is consistent with the critical and compelling intent of the author.
First, while the application of historical cases/data is compelling, since the author is describing racism, structural racism in particular, establishing parallels with some of the work that shows the weathering effects of racism today will help strengthen the argument. Additionally, at the conclusion, one of the broader implications from the authors argument is the need to rethink of data in ways that allow social scientist studying structural racism and health disparities to integrate historical data to contemporary racialized structures and policies to help show the persistence of white supremacy through time and space.
Finally, the paper needs thorough editing. There are several areas of grammatical errors. Below are some of the examples of some of these errors.
The abstract the second line is to not of
Based on four decades of empirical research, Geronimus posits that.
On line 5, it should be either results in accelerated decline or declines.
On page 2, line 6 of the second paragraph should read the impact of structural racism on the marginalized .
On Line 10 of the same paragraph, there should be a coma (,) after the structural racism and violence [before] individual.
On page 3, line 18 under the second paragraph needs to be of, not row, racial oppression. On the same line, results prolonged elevated, should be result in prolonged.
On page 7, line 2 of the paragraph before the conclusion, the word weekend should be checked. It appears to be weakened, but it reads as weekend.
Comments on the Quality of English Language
Overall, this article drawing on the experiences the Jews forcefully concentrated into Łódź ghetto from 1939-1945, argues that the theory of weathering holds broader explanatory power that can help unpack the violence inherent in racialized structures, policies, and practices and that produces avoidable death through multiple pathways: physical health, premature aging, psychological distress, family breakdowns, and many more. The argument put forth by the author is very compelling. However, I think the paper can be strengthened to bring the paper to the level that is consistent with the critical and compelling intent of the author.
First, while the application of historical cases/data is compelling, since the author is describing racism, structural racism in particular, establishing parallels with some of the work that shows the weathering effects of racism today will help strengthen the argument. Additionally, at the conclusion, one of the broader implications from the authors argument is the need to rethink of data in ways that allow social scientist studying structural racism and health disparities to integrate historical data to contemporary racialized structures and policies to help show the persistence of white supremacy through time and space.
Finally, the paper needs thorough editing. There are several areas of grammatical errors. Below are some of the examples of some of these errors.
The abstract the second line is to not of
Based on four decades of empirical research, Geronimus posits that.
On line 5, it should be either results in accelerated decline or declines.
On page 2, line 6 of the second paragraph should read the impact of structural racism on the marginalized .
On Line 10 of the same paragraph, there should be a coma (,) after the structural racism and violence [before] individual.
On page 3, line 18 under the second paragraph needs to be of, not row, racial oppression. On the same line, results prolonged elevated, should be result in prolonged.
On page 7, line 2 of the paragraph before the conclusion, the word weekend should be checked. It appears to be weakened, but it reads as weekend.
Author Response
I appreciate your close and careful reading of the manuscript. I have reworked with article to address your critiques. I have tried to make the connection between racist/genocidal Nazi policies and the phenomenon of weathering in this context clearer throughout the work. The latter portion of the manuscript was revised to clarify how communal weathering manifested itself in the Lodz ghetto and added some additional references. I also corrected the many spelling and grammatical errors found in the first iteration of the manuscript.
Reviewer 2 Report
Comments and Suggestions for Authors
The article ““Everything Is Old”: National Socialism and the Weathering of the Jews of Łódź” proposes a broadening of the application of the term “weathering” due to sociologist Arline Geronimus – who used it to explain the reason for health deterioration and premature aging of Blacks in comparison to whites in the U.S. as a result of racism – to the case of Jews in the Łódź ghetto because of Nazi policies. In showing how the weathering of the Jews of Łódź happened because of hunger, fear, and threats of violence posed “by a society governed by racist systems and practices” (see page 10 line 472), the article offers a well-rounded starting point for a thorough investigation of Jews’ individual deterioration leading to a deterioration of the Jewish community as a whole in the Łódź ghetto, to be developed further by scholars.
To prove the argument, the author has divided the article into four rigorously chosen and interconnected sections. What needs to be added is an explanation of how the author connects the “weathering” concept about African Americans’ discrimination and physical and psychological deterioration because of white supremacy in the U.S. with the “Nationalist Socialist white supremacy” (page 2 line 93-94) behind Nazi treatment of Jews in Łódź and other areas during the Holocaust. In this sense, the author could explain the Nationalist Socialist white supremacy in relation to Nazis’ development of eugenics under which Jews were treated and connect this to the use of eugenics for the discrimination of African Americans in the U.S. Furthermore, some previous scholarship linking the treatment of Jews in the Holocaust and African Americans at various points in time should also be integrated in the article’s discussion. This can include, for instance, references to Danielle Christmas’s works to date: i.e. “The Plantation-Auschwitz Tradition: Forced Labor and Free Markets in the Novels of William Styron.” Twentieth-Century Literature 61.1 (Spring 2015): 1-31, or Danielle Christmas and Adam Brown. “When the Holocaust Comes to Harlem: Traumatic Memory, Race, and Economic (In)Justice in American Holocaust Film.” In Mapping Generations of Traumatic Memory in American Narratives, 2014.
In the conclusion, I think the author should also add a paragraph about the similar and different elements of weathering in the case of Jews in Łódź ghetto and African Americans in the U.S. previously discussed by sociologists, indicating what sets the two cases apart as to official policies (e.g. systematic exploitation and extermination in the case of the Nazis, systematic exploitation in the case of the U.S.), but also physical and psychological dimensions of weathering based on the author’s findings.
On page 2, lines 58-59, a footnote including the sources behind the numbers of Łódź Jews in the period 1940-1945 should be added.
Please also delete the last part of the paragraph before the conclusion starting with “Death became a routine part of life” – lines 351 to lines 358. The same part of the paragraph has already been included on page 6 lines 298-307. Overall, I think the article greatly fits into the scope of Genealogy; I recommend it for publication in Genealogy after the author makes the minor corrections I suggested above.
Comments on the Quality of English Language
Please carefully re-read the entire article again, as it contains quite a lot of typos. Below is a list of the typos that caught my eye and need correction:
Page 1 line 5 replace “of describe” by “to describe”
Page 1 line 8 replace “accelerate declines” by “accelerated declines”
Page 1 line 34 delete “he” after from
Page 1 Line 39 delete one “and”
Page 2 Line 50 delete one “and”
Page 2 line 58 replace “impact structural racism” by “impact of structural racism”
Page 2 line 68 replace “Based more” by “Based on more”
Page 2 Line 81 replace “row racial oppression results prolonged” by “of racial oppression results in prolonged”
Page 2 line 89 replace “on the theory” by “of the theory”
Page 3 line 110 replace “jews” by “Jews”
Page 3 line 122 replace “weathering the on vitality” by “weathering on the vitality”
Page 3 line 122 replace “I argue prolonged” by “I argue that prolonged”
Page 3 line 139 replace “German authorities early weeks” by “German authorities since the early weeks”
Page 3 line 141 delete one comma of dispossession
Page 3 line 152 delete “of” after “descriptions” and one of the two uses of “of” before “a faceless mass”
Page 4 line 154 The verb is missing for the sentence that starts on this line; it should read “Itzhak Cytrybowski described the impact”
Page 4 line 168 replace “regime to relentless terror” by “regime of relentless terror”
Page 4 line 168 – check the sentence starting with “n Lodz erected a fence….” – clearly, something is missing here and the sentence needs rewriting.
Page 4 line 171 replace “at the ghetto borden” by “at the ghetto border”
Page 4 line 173 replace “percarity” by “precarity”
Page 4 line 189 replace “authorities bestow” by “authorities bestowed”
Page 4 line 199 replace “effects daily depravation and injustice” by “effects of daily depravation and injustice”
Page 5 line 219 replace “state of sanitation the ghetto” by “state of sanitation of the ghetto”
Page 5 line 221 replace “accelerate weathering” by “accelerated weathering”
Page 6 line 296 delete either “dimensions” or “components”
Page 6 line 299 replace “accelerate aging” by “accelerated aging”
Page 6 line 299 replace “no mater there age” by “no matter their age”
Page 6 line 300 replace “remember” by “remembered”
Page 7 line 318 replace “with in the native community” by “within the native community”
Page 7 line 327 delete “of” after “Mortality rates”
Page 7 line 345 replace “the bones natural defenses” by “the bones’ natural defenses”
Page 7 line 351 replace “accelerate aging” by “accelerated aging”
Page 7 line 362 delete one “the” before “individual”
Page 9 line 463 replace “on an in individual level” by “on an individual level”
Page 9 line 468 to page 10 line 470 – rewrite the sentences – they are incorrect grammatically
Page 10 line 472 replace “the impact living” by “the impact of living”
Page 10 line 477 replace “feature of the weathering process” by “features of the weathering process”
Page 10 reference 3 line 490 replace “based four decades” by “based on four decades” and “Unite States” by “United States”
Page 10 line 521 replace “emphasized the need” by “emphasizing the need”
Author Response
I appreciate your close and careful reading of the manuscript. I have reworked the article to address your critiques. I have tried to make the connection between racist/genocidal Nazi policies and the phenomenon of weathering in this context clearer throughout the work. The latter portion of the manuscript was revised to clarify how communal weathering manifested itself in the Lodz ghetto and added some additional references. I appreciate your suggestions about making more explicit the parallels between this historical case and contemporary examples of weathering and its effects. I have tried to integrate appropriate references to highlight this parallel and suggest that more work could be done here. I also corrected the many spelling and grammatical errors found in the first iteration of the manuscript.
Round 2
Reviewer 2 Report
Comments and Suggestions for Authors
The author has thoroughly integrated my suggestion to explain the Nationalist Socialist white supremacy in relation to Nazis' development of eugenics and to the use of eugenics for the discrimination of African Americans in the U.S. Its proposed broadening of using the concept of "weathering" in the case of Jews in the Lodz ghetto is very soundly and ethically explained and placed in relevant conversation with previous scholarly studies. I fully recommend the article for publication in Genealogy and look forward to seeing it published.